# DNA Methylation Signatures of Multiple Sclerosis Occur Independently of Known Genetic Risk and Are Primarily Attributed to B Cells and Monocytes

**DOI:** 10.3390/ijms241612576

**Published:** 2023-08-08

**Authors:** Alexandre Xavier, Vicki E. Maltby, Ewoud Ewing, Maria Pia Campagna, Sean M. Burnard, Jesper N. Tegner, Mark Slee, Helmut Butzkueven, Ingrid Kockum, Lara Kular, Vilija G. Jokubaitis, Trevor Kilpatrick, Lars Alfredsson, Maja Jagodic, Anne-Louise Ponsonby, Bruce V. Taylor, Rodney J. Scott, Rodney A. Lea, Jeannette Lechner-Scott

**Affiliations:** 1School of Biomedical Sciences and Pharmacy, Hunter Medical Research Institute, University of Newcastle, New Lambton Heights, NSW 2305, Australia; alexandre.xavier@newcastle.edu.au (A.X.); sean.burnard@newcastle.edu.au (S.M.B.); rodney.scott@newcastle.edu.au (R.J.S.); 2School of Medicine and Public Health, Hunter Medical Research Institute, University of Newcastle, New Lambton Heights, NSW 2305, Australia; vicki.e.maltby@newcastle.edu.au (V.E.M.); rodney.lea@qut.edu.au (R.A.L.); 3Department of Neurology, John Hunter Hospital, New Lambton Heights, NSW 2305, Australia; 4Department of Clinical Neuroscience, Karolinska Institutet, Center for Molecular Medicine, Karolinska University Hospital, 17176 Stockholm, Sweden; ewoud.ewing@ki.se (E.E.); ingrid.kockum@ki.se (I.K.); lara.kular@ki.se (L.K.); lars.alfredsson@ki.se (L.A.); maja.jagodic@ki.se (M.J.); 5Department of Neuroscience, Central Clinical School, Monash University, Melbourne, VIC 3004, Australia; maria.campagna@monash.edu (M.P.C.); helmut.butzkueven@monash.edu (H.B.); vilija.jokubaitis@monash.edu (V.G.J.); 6Biological and Environmental Science and Engineering Division, King Abdullah University of Science and Technology (KAUST), Thuwal 23955-6900, Saudi Arabia; jesper.tegner@kaust.edu.sa; 7Computer, Electrical and Mathematical Sciences and Engineering Division, King Abdullah University of Science and Technology (KAUST), Thuwal 23955-6900, Saudi Arabia; 8Unit of Computational Medicine, Department of Medicine, Center for Molecular Medicine, Karolinska Institutet, Karolinska University Hospital, L8:05, 17176 Stockholm, Sweden; 9Science for Life Laboratory, Tomtebodavagen 23A, 17165 Solna, Sweden; 10College of Medicine and Public Health, Flinders University, Bedford Park, SA 5042, Australia; mark.slee@flinders.edu.au; 11MSBase Foundation, Melbourne, VIC 3004, Australia; 12Florey Institute of Neuroscience and Mental Health, The University of Melbourne, Melbourne, VIC 3052, Australia; tkilpat@unimelb.edu.au (T.K.); annelouise.ponsonby@florey.edu.au (A.-L.P.); 13National Centre for Epidemiology and Public Health, Australian National University, Canberra, ACT 2601, Australia; 14Menzies Institute for Medical Research, University of Tasmania, Hobart, TAS 7000, Australia; bruce.taylor@utas.edu.au; 15Department of Molecular Genetics, Pathology North, John Hunter Hospital, New Lambton Heights, NSW 2305, Australia; 16Centre for Genomics and Personalised Health, School of Biomedical Science, Queensland University of Technology, Kelvin Grove, QLD 4059, Australia

**Keywords:** multiple sclerosis, epigenetics, methylation, epigenome-wide association studies, genetic risk, cell deconvolution

## Abstract

Epigenetic mechanisms can regulate how DNA is expressed independently of sequence and are known to be associated with various diseases. Among those epigenetic mechanisms, DNA methylation (DNAm) is influenced by genotype and the environment, making it an important molecular interface for studying disease etiology and progression. In this study, we examined the whole blood DNA methylation profiles of a large group of people with (pw) multiple sclerosis (MS) compared to those of controls. We reveal that methylation differences in pwMS occur independently of known genetic risk loci and show that they more strongly differentiate disease (AUC = 0.85, 95% CI 0.82–0.89, *p* = 1.22 × 10^−29^) than known genetic risk loci (AUC = 0.72, 95% CI: 0.66–0.76, *p* = 9.07 × 10^−17^). We also show that methylation differences in MS occur predominantly in B cells and monocytes and indicate the involvement of cell-specific biological pathways. Overall, this study comprehensively characterizes the immune cell-specific epigenetic architecture of MS.

## 1. Introduction

Multiple sclerosis (MS) is a chronic autoimmune and neurodegenerative disease of the central nervous system (CNS). It is caused by the combination of a complex genetic predisposition and environmental and lifestyle factors. Genome-wide association studies (GWAS) have identified over 200 single nucleotide polymorphisms (SNPs) associated with MS risk, the majority of which localize to genes relevant to immune function [1,2]. The strongest risk locus is found within the major histocompatibility complex (MHC) region of chromosome 6—the well characterized *HLA-DRB1***1501* MS risk allele [3,4,5].

Despite the progress made by GWAS, the exact pathogenesis of MS remains to be elucidated. There is good evidence for dysregulation of immune cell balance in the CNS, particularly with regards to immune phenotypes [6,7]. Studies suggest that altered ratios between adaptive and innate immune system cells lead to disease activity [6,7]. There are also a number of well-established environmental and lifestyle factors that contribute to disease risk, such as viral infection (specifically Epstein Barr Virus (EBV) and infectious mononucleosis [8,9,10]) and smoking [11], all of which interact with the genome. Other factors that have no direct genomic influence but are considered to contribute to disease include low ultraviolet radiation exposure, low vitamin D levels [12], and nulliparity [13]. The exact molecular mechanisms by which MS develops remain elusive, but epigenetic change is emerging as a contender that significantly influences disease risk.

One of the most well-studied epigenetic modifications is DNA methylation (DNAm), where increased levels of CpG methylation are generally associated with gene silencing and demethylation with gene activation. Epigenetic modulation via CpG methylation is unique in reflecting environmental exposure and lifestyle factors, making it a plausible modifier of MS onset and progression. Epigenome-wide association studies (EWAS) of DNA have identified changes in DNA methylation profiles associated with MS [14,15,16]. Although these studies have reported somewhat conflicting results, they all consistently identified differential methylation at *HLA-DRB1***1501*, which appears to partly mediate the relationship between the risk allele at this locus and MS [16]. The limited data from isolated cell type analyses suggest that each immune cell population has a unique epigenetic profile [17,18,19,20]. To date, isolated cell type EWAS studies have involved modest sample sizes, whereas larger whole blood EWAS studies have likely missed cell type-specific differential methylation due to masking of the signals from mixed cell populations. Statistical methods now exist to estimate the relative proportions of major blood cell types using methylation data derived from mixed cell DNA samples (reviewed by Campagna et al. [21]). Methods to identify cell-specific differential methylation (csDNAm) are derived from these algorithms (CellDMC).

In this study, we have conducted an EWAS of whole blood DNA from MS cases and controls and applied statistical deconvolution to reveal cell-specific methylation signatures associated with the disease. Importantly, the discovery group (from the Ausimmune study) included cases with early MS (principally first clinical diagnosis of CNS demyelination) who had a follow-up assessment of disease after 10 years [22]. This allowed us to identify causative epigenetic changes in specific cell subsets that occurred prior to, or early in, disease onset rather than as a consequence of long-standing pathology or therapeutic intervention. Validation analysis confirmed our results in two independent studies.

## 2. Results

The study design is described in Figure 1 and in the methods below.

### 2.1. DNA Methylation Analysis Implicates the HLA Locus

The strongest methylation differences were identified within the major histocompatibility complex (MHC) class II locus, within HLA genes. This replicated the findings of previous methylation studies in MS and highlighted again the importance of the MHC locus in MS. We first identified 13 replicated differentially methylated positions (DMPs) (see Appendix A), with 4 DMPs closely linked to interferon-related pathways (see Appendix A).

The combined dataset analysis identified 11,969 individual DMPs that met the genome-wide significance threshold of *p* < 9.8 × 10^−8^. Of these, 190 also had Δβ > 0.02 and 81 were located at the MHC class II locus (see Figure 2a). Differentially methylated region (DMR) analysis identified 9 major DMRs across HLA-D genes (see Appendix A).

We then performed mediation analysis using the causal inference test (CIT) method [23] to test for causality between the *HLA-DRB1* risk haplotype (casual factor), DMR methylation (mediator), and clinical definite MS (CDMS) as the phenotype. CIT analysis allowed the establishment of a chain of causality between genotype, methylation, and MS. The results of the CIT analysis showed that the methylation differences identified between pwMS and controls were driven entirely by genotype in both the discovery and first validation groups (*p* < 0.05) (see Figure 2b) (the second validation group did not have genotyping information).

To determine if methylation was correlated with gene expression, we examined the relationship between methylation and gene expression using RNA extracted from isolated monocytes, CD19^+^ B cells, and CD4^+^ T cells from our previously published study [16], one unpublished study, and an unpublished dataset (Maltby et al., unpublished) (see methods for group details). We used Pearson’s correlation to estimate the relationship between methylation and expression levels of *HLA-DRB1* in all three cell types (see Figure 2c) and found a strong negative correlation between gene expression and methylation at DMR-2 located in exon 2 of *HLA-DRB1.* These data indicated that hypomethylation at this locus was associated with increased gene expression in these immune cell subtypes. As expected, based on the DNA methylation levels, there was a strong correlation between *HLA-DRB1* expression and methylation levels, but this correlation was not as strong in CD4^+^ T cells.

### 2.2. Methylation Differences Occur Independently of Known Genetic Risk

Given the strong correlation between methylation and genotype at the MHC region (see Section 2.1 and Appendix A), we wanted to determine if epigenetic risk was independent of the known genetic risk of MS. To do this, we computed EWAS models controlling for the effect of both the *HLA-DRB1* risk haplotype and non-MHC SNPs identified in the 2019 International Multiple Sclerosis Genetics Consortium GWAS [2] using a polygenic risk score (PRS). This PRS was calculated using plink files with PRSice2 [24]. This analysis was only performed in the discovery group because (a) there were missing genetic data from the validation data sets and (b) the discovery group represented early-stage MS, which is an important attribute for risk assessment. From this analysis, we identified 1627 DMPs reaching statistical significance (*p* < 9.8 × 10^−8^), 123 of which had Δβ above 2% (see Appendix A). None of the DMPs localized to the *HLA-DRB1* gene, which suggested that the epigenetic effects identified at this locus were linked to genetic variation. To assess if the identified DMPs were modulated by genotype, we tested each CpG for cis-acting methQTLs. An absence of metQTLs suggested that the methylation differences were most likely the result of lifestyle and/or environmental exposure rather than genetic influence. Only 104 out of 1627 (6.4%) produced a significant association with an SNP genotype, showing evidence of modulation by a nearby genetic locus (see Figure 3a). Similarly, we tested each CpG for association with each SNP that was used to construct the PRS. A total of 98 out of 1627 CpGs (6.02%) showed significant modulation by any PRS-associated SNP. The 1627 DMPs mapped to a total of 1092 unique genes. To explore potentially affected biological pathways, we performed over-representation analysis (ORA) on all genes containing DMPs. The list of 1092 unique genes identified after genotype correction revealed emerging pathways in MS such as the *NOTCH* signaling and axon guidance pathways (see Figure 3b).

Using these DMPs, we then constructed a methylation risk score that was corrected for all known genetic risk, both inside and outside of the MHC region (grcMethScore) for each subject. This grcMethScore represents the epigenetic risk of MS independent of the known genetic risk. We then compared the grcMethScore to the *HLA-DRB1* risk haplotype and PRS to assess how well each feature distinguished pwMS from controls. We found that the PRS and *HLA-DRB1* risk haplotype were not correlated (R = 0.067, 95% CI: −0.013–0.14, *p* = 0.1), indicating that the genetic risk conferred by *HLA-DRB1* was independent of other genetic risk factors. Next, we constructed a total known genetic risk score, which incorporated both the PRS and *HLA-DRB1* risk haplotype. The grcMethScore only had a very weak correlation with total genetic risk (see Figure 3c) (R = 0.23, 95% CI: 0.15–0.30, *p* = 8.2 × 10^−9^).

To assess how our three risk scores (PRS, HLA-DRB1*15:01, and grcMethScore) could act as classifiers for MS, we conducted area under the curve (AUC) receiver operating curve (ROC) analysis (see Figure 3d). We showed that the total known genetic risk score (HLA-DRB1 + PRS) provided moderate discriminatory accuracy (AUC = 0.72, 95% CI: 0.66–0.76). By comparison, the grcMethScore performed better at distinguishing MS from controls (AUC = 0.85, 95% CI: 0.82–0.89). Combining genotype with methylation score only marginally improved the accuracy of the grcMethScore (AUC = 0.87, 95% CI: 0.84–0.91), as did the addition of all other covariates, such as cell proportions, sex, and age (AUC = 0.89, 95% CI: 0.86–0.92).

### 2.3. Cell-Specific Differential Methylation Is Mainly Attributed to B Cells and Monocytes

To identify if specific immune cell types were responsible for the differential methylation identified in the whole blood samples, we employed a two-step approach using the combined dataset. First, we calculated cell proportion estimates in the combined dataset using the R package EpiDISH [24]. This revealed a statistically significant decrease in the proportions of natural killer (NK) cells (mean_case_ = 0.021, mean_control_ = 0.031, *p* = 1.34 × 10^−8^) and CD8^+^ T cells (mean_case_ = 0.095, mean_control_ = 0.106, *p* = 5.92 × 10^−5^) and a statistically significant increase in the CD4^+^ T cell proportions (mean_case_ = 0.081, mean_control_ = 0.071, *p* = 1.58 × 10^−3^) in MS cases compared to controls. There were no differences in the proportions of monocytes, B cells, and neutrophils between cases and controls (Appendix A).

Second, we used these cell type proportions, in conjunction with CpG methylation (M) values, to calculate cell type-specific DMPs (csDMPs). Here we employed an adaptation of the functions in CellDMC [25]. Although it accounts for all cell types simultaneously, the base CellDMC regression model can become overburdened when incorporating many cell types (i.e., many terms), thus reducing the power and preventing the identification of cell-specific effects. To overcome this, we constructed a “per cell type” linear model (see methods Section 4).

Genome-wide analysis of cell-specific methylation patterns revealed an over-representation of csDMPs in both B cells and monocytes. From the highest number of csDMPs to the lowest were: B cells > monocytes > CD4^+^ T cells > CD8^+^ T cells > NK cells, and then neutrophils (Figure 4a). Interestingly, csDMPs were predominantly hypermethylated in B cells (over 60%) while monocytes had a more evenly distributed set of csDMPs (Figure 4a). Consistent with the whole blood analysis, there were marked differential methylation effects at *HLA*, but notably, this effect exhibited obvious variation across cell subtypes (Figure 4b). More detailed analysis of the cell-specific modeling revealed that the strong differential signal observed at the *HLA* region in whole blood originated primarily from monocytes and B cells but was also present to a lesser extent in T lymphocytes (Figure 4). It was notably absent or nearly absent in both NK cells and neutrophils (Appendix A).

Comparison of the four cell types with the highest number of csDMPs that met both the significance and effect size cut-offs (monocytes, B cells, CD4^+^ T cells, and CD8^+^ T cells) revealed that both B cells and monocytes had largely different sets of csDMPs. There were 1165 CpGs unique to B cells, 771 CpGs unique to monocytes, and 347 csDMPs common to both. When we investigated all cell types, we found that CD4+ T cells, CD8+ T cells, monocytes, and B cells shared 28 common csDMPs, with the same directionality. These were located on 7 genes: *HLA-DQB1*, *CMPK2*, *IGF2R*, *HLA-DRB1*, *HLA-DRB6*, *HLA-DRB5*, and *HLA-DQA1* (Appendix A and Appendix A).

Our validation groups had older mean ages and longer disease duration than the discovery group. To examine if the predominant methylation signals coming from B cells and monocytes occurred early in disease pathology or was a result of long-standing disease, we performed the same analysis described above in only the discovery group. As in the combined analysis, we found the majority of DMPs appeared to be from B cells and monocytes, to a lesser extent from T lymphocytes, with absent or nearly absent signals from both NK cells and neutrophils (see Appendix A).

### 2.4. Over-Representation Analysis Shows Enrichment of DMPs in Immunological Pathways

In the combined whole blood group, we identified 190 DMPs with Δβ > 2%. These DMPs mapped to 92 unique genes (Appendix A). Over-representation analysis (ORA) using this gene list identified 28 pathways. The ORA showed enrichment in immunological pathways, including cytokine- and interferon-related pathways as well as antigen presentation through MHC class I and II (Appendix A).

Our cell-specific analysis identified 1740 csDMPs in B cells and 1212 csDMPs in monocytes (Figure 2a). We created two gene lists, one for B cells and one for monocytes, using genes associated with at least one csDMP from that cell type. We then used the two gene lists as well as the gene list from the whole blood DMP analysis to perform over-representation analysis. All the pathways identified in B cells were also present in both whole blood and monocytes. However, the gene list from monocytes was associated with pathways not found in B cells. For example, the cytokine signaling and IFN alpha/beta pathways identified in whole blood were only found in monocytes and not in B cells, suggesting cell-specific changes (Figure 5).

## 3. Discussion

Despite the significant progress in developing treatments for MS, the pathogenic molecular events that occur prior to, and early in, the disease are not understood. The combined effects of genetics and environmental risk factors suggest that epigenetic mechanisms are likely to be involved.

The key results of this study, recapitulated in Figure 6, indicated that methylation is a better discriminator of MS status than genetics. The moderate discriminatory power (risk) accuracy emerging from both the PRS and *HLA-DRB1* haplotype underlines the limited role played by genetics in MS risk [2]. Correcting for known strong MS risk loci, we identified robust CpGs that were largely not modulated by genotype. We provided evidence that methylation alone is a stronger discriminator of MS than previously described known genetic risk factors. A chain of causation still needs to be established, identifying if these changes in methylation are causative or a consequence of MS. Since these findings occurred in an early-stage MS group (most at the first demyelinating event (FDE)), we can rule out that these effects were a consequence of chronic disease. However, whether these effects occur at birth, during the prodromal period, or at disease onset will be difficult to determine.

Since DNA methylation is substantially influenced by environmental factors, the results highlight the importance of environmental exposure in MS risk. Further studies need to systematically evaluate the effect of environmental exposure on methylation change and MS risk, such as seen in vitamin D [26], smoking [27], or EBV infection. Interestingly, controlling for the influence of genotype also unveiled non-immune-related pathways disrupted in pwMS, such as the NOTCH signaling or axon guidance pathways, both of which emerged as key pathways in MS [28,29]. Axon guidance molecules are known to modulate axon regeneration in disease and are involved in different inflammation-related pathways [30]. NOTCH signaling regulates oligodendrocyte progenitor cell differentiation and downstream myelin production [31].

Numerous studies have now been published investigating the genome-wide DNA methylation changes in whole blood and isolated cell types; however, the results are inconsistent and limited by heterogeneous cell populations, disease heterogeneity, use of a wide variety of disease modifying therapies (DMTs), or small sample size [14,16,18,19,32]. Using a large dataset of MS cases early in their diagnosis with age/sex/location-matched controls, we confirmed previous findings that DNA methylation acts as mediator of the *HLA-DRB1***1501* allele [16] and provided evidence that this occurs early in disease pathology rather than as a result of treatment or long-standing disease. Furthermore, we revealed that the differential methylation was primarily associated with B cells and monocytes, with a smaller contribution from T lymphocytes (CD4^+^ and CD8^+^). This is consistent with our previous observations in these four isolated cell types [18]. We also correlated methylation change with gene expression at the *HLA-DRB1* locus, thereby substantiating its key role in MS pathogenesis, which we replicated in two independent patient cohorts. However, outside of the MHC region, significant methylation changes appeared to be unrelated to meaningful genotype changes thought to modulate MS risk (identified through the 2019 IMSGC GWAS [2]).

A major strength of our study was the discovery group, with data collected early in diagnosis and the group followed prospectively over a ten-year period. This allowed us to examine the methylation profiles of pwMS at baseline collection, who, at their 10-year follow-up visit, had fulfilled the criteria for MS. We demonstrated that B cells and monocytes accounted for the largest contribution of the whole blood DNA methylation signal in MS and that this was not simply a result of long-standing disease. This is consistent with previous work on isolated cell subtypes from pwMS [17,18,33] and with current knowledge of B cells playing an important role in the pathogenesis of disease, highlighted by the efficacy of treatments targeting the CD20 surface antigen predominantly expressed on B cells [34]. There has been the suggestion that these therapies specifically target the memory B cell compartment [35], so future studies investigating the methylation of memory B cells will be of interest. In contrast to previous work, the csDMPs identified in B cells were predominantly hypermethylated.

Monocytes, which also have antigen-presenting properties, likely play an important role in early-stage disease by mediating both pro- and anti-inflammatory responses [36,37,38]. Furthermore, our ORA and GAT analyses revealed monocyte differential methylation signals included in unique pathways, suggesting they may play an early role in MS pathogenesis. Increased proportions of CD4^+^ T cells and a deficit of CD8^+^ T cells in MS cases compared to controls have been previously documented [39,40,41]. The increased proportions of CD4^+^ T cells are likely driven by Th17 helper cells, whereas the deficit of CD8^+^ T cells is probably due to reduced effector cell numbers [39,40,41]. Our cell proportion estimates are consistent with these past findings. The proportion of total NK cells has not previously been evaluated in MS; however, we have previously demonstrated that subsets of NK cells are affected by disease-modifying therapies and an increase in the abundance of NK_bright_ cells is associated with stable disease [42]. To overcome issues of heterogeneous cell populations and small sample size, we used an adapted CellDMC method to identify which cell types contributed the most to the differential methylation signal. We were able to demonstrate that the main source of differential methylation in whole blood came from B cells and monocytes, particularly in the *HLA-DR* region. This is not surprising, given that HLA-DR is expressed primarily on antigen-presenting cells such as B lymphocytes and monocytes [43]. We also found strong differential methylation signals from both CD4^+^ and CD8^+^ T cells. HLA-DR is normally absent on resting T lymphocytes; however, it has been proposed as a marker of immune system activation [44]. In systemic lupus erythematosus, CD8^+^ T and CD4^+^ T cells have increased HLA-DR expression during active disease, leading to speculation that expression of HLA-DR represents an inflammatory response. It is plausible that this is also happening in the T lymphocytes of people with MS, but further studies are required to confirm this.

In addition to the 13 DMPs located at *HLA* genes, we also identified two robust signals that localized to interferon-related genes. This suggests that these DMPs may be partially influenced by treatment. One small study found global methylation changes in monocytes after treatment with interferon beta [45]. The authors of this study, however, were unable to identify changes at specific CpGs [45]. A study of monozygotic twin pairs with MS found DMPs associated with IFN beta treatment; however, we did not find any overlap with this study [46]. A larger sample size and prospective study design to confirm these findings is required. The results of our csDMP analysis was compared to two previously reported studies, Ewing et al. [18] and Ma et al. [33], which revealed 154 shared genes containing csDMPs in B cells and two shared genes in monocytes in all three studies (Appendix A), suggesting these may be very robust DMPs for these cell types. No shared DMPs were identified in either the CD4^+^ or CD8^+^ T cell populations. Differences would be expected between studies due to the use of different assays (Ewing et al. used the 450 K platform and Ma et al. used bisulfite sequencing), different group statistical powers, and different group characteristics. The overlap likely represents replicated MS-associated DMPs.

One limitation of this study is that the combined dataset was heterogenous in terms of clinical characteristics. Although our discovery group was primarily treatment naïve and early in their disease course, the validation groups contained participants on mixed treatments and with long-standing disease. Thus, we cannot draw conclusions regarding how treatment use or disease progression affects methylation change. Furthermore, despite the large size of the combined dataset, the statistical deconvolution methods were still only likely to identify major effect DMPs. Smaller effect CpGs probably exist, but much larger studies are needed to identify them. Another limitation is that the expression data were from separate participant samples; therefore, we cannot assign causation between expression and methylation, we can only make correlations. Future studies with larger groups of treatment-naïve populations, followed prospectively over time, are necessary to confirm our DNA methylation and expression data and confirm the results presented herein. Finally, we acknowledge that ORA is an exploratory analysis to identify potentially affected biological pathways and help guide further studies, but the results need to be confirmed at a molecular level.

## 4. Materials and Methods

See Figure 1 for a study workflow diagram.

Clinical characteristics have been listed in Appendix A.

### 4.1. Whole Blood Methylation

The discovery group was comprised of whole blood samples collected at baseline from The Australian Multi-Centre Study of Environment and Immune Function (Ausimmune study) [22]. From this group, there were 208 MS cases and 402 matched non-MS controls. Most of this group entered the Ausimmune study after their first clinical diagnosis of CNS demyelination (FCD), were primarily treatment naïve at the time of sampling and had mainly the relapse-onset phenotype. We used whole blood samples from the baseline collection in cases who, at the 10-year follow-up, had converted to MS based on the 2017 McDonald criteria [47]. Controls were matched for age, sex, and region [22].

The first validation group was from Sweden and consisted of 140 MS cases and 139 controls. This dataset consisted of previously published data from prevalent MS cases who were primarily on treatment at the time of blood sampling, mostly had the relapse-onset phenotype, and had been profiled using the Illumina 450 K array [16].

The second validation group consisted of 235 MS cases sourced from four Australian MS specialist clinics in Melbourne, Newcastle, Hobart, and Adelaide. The cases in this group exclusively had the relapse-remitting phenotype, but had longer disease duration (median 16 years), and were on a variety of treatments at sample collection [48]. This case group did not have controls, so they were matched with data from 102 non-MS controls from a publicly available dataset.

All DNA samples from the discovery groups and case samples from the second validation group were tested in-house for genome-wide methylation using the Illumina Infinium Human Methylation EPIC BeadChip Kit (EPIC array) (Illumina Inc., San Diego, CA, USA). The publicly available dataset matched to the second validation group was also profiled using the EPIC array.

To identify DNA methylation changes associated with MS, we performed an EWAS on bisulfite-treated genomic DNA isolated from whole blood (see Figure 1 and methods Section 4). A total of 583 people with MS (pwMS) and 643 controls from three independent groups were analyzed in this study. Further details of each group are shown in Appendix A and in the methods section.

Discovery Group: details of the Ausimmune Study design have been previously published [22]. Briefly, between 2003 and 2006, participants were recruited across four different centers in Brisbane, Newcastle, and surrounding areas; Geelong; the Western Districts of Victoria; and the Island of Tasmania. Cases were aged 18–59 years and entered the study around the time of their first demyelinating event (FDE). They were prospectively followed over 10 years. The Australian electoral roll was used to select controls from the general population at baseline. Non-MS controls were matched based on age (±2 years), gender, and region, with a ratio of up to 4 controls for each case.

Epidemiological information was collected by face-to-face interview, self-completed questionnaires, and neurological examination. At the same time, biological samples were collected and stored. Detailed analysis of the information collected has been previously published [22]. For this study, we had access to whole blood from 208 MS cases and 402 matched controls. All 208 MS cases had converted to CDMS by the 10-year follow-up.

The first validation group was from a previously published case–control study (group 2) and the 450 K array data were available from the Gene Expression Omnibus (GEO) database under accession numbers GSE43976 and GSE106648. The dataset was comprised of whole blood DNA from 140 MS cases and 139 non-MS controls. Samples were collected in Sweden between 2005 and 2011, the cases were mostly on low-efficacy treatment (65%) and mostly had the relapsing-remitting phenotype.

The second validation group was comprised of 235 MS cases obtained from the specimen banks of 4 Australian specialist centers in Melbourne, Newcastle, Hobart, and Adelaide and 102 publicly available control idat files obtained through the EWAS data hub (https://ngdc.cncb.ac.cn/ewas/datahub/repository, accessed on 8 July 2022) using the following criteria: age between 23 and 73 years, sex = female, platform = 850 K, tissue = whole blood, and sample type = control. The second validation group was publicly available in the GEO database (GSE106648) and was previously used in a whole blood case–control EWAS [48].

### 4.2. DNA Extraction and Quality Control

For the discovery group and second validation dataset, DNA was extracted from whole blood using the QIAamp DNA Blood Mini Kit™ (Qiagen, Venlo, The Netherlands,). Extracted DNA was quantified using the Qbit (Invitrogen, Waltham, MA, USA) and DNA integrity was assessed using the Genomic DNA ScreenTape assay with the TapeStation system (Agilent, Santa Clara, CA, USA), using the DNA integrity number (DIN) as a metric. All samples had a DIN ≥ 7, indicating minimal genomic DNA degradation.

### 4.3. Methylation and Genotyping Arrays

To determine the SNP profiles, the Global Screening Array v2.1 (Illumina, USA) (hereafter referred to as GSA) was used according to the manufacturer’s guidelines. For genome-wide methylation analysis, 500 ng of genomic DNA was bisulfite-converted using the EZ-DNA Methylation™ Kit (Zymo Research, Irvine, CA, USA,) according to the manufacturer’s guidelines. The converted DNA was hybridized to the Illumina Methylation EPIC BeadChip array (hereafter referred to as EPIC array). To avoid batch effects, samples were randomized on both the GSA and EPIC arrays using the R package OSAT [49]. Both the GSA and EPIC arrays were read using the iScan system (Illumina Inc., San Diego, CA, USA) to produce raw idat files.

### 4.4. Methylation Analysis

For the discovery and both validation groups, analysis was performed using the R package ChAMP [50,51]. To summarize, idat files were loaded and filtered to remove low performing probes, probes mapping to multiples loci, and low performing samples. Probes next to known polymorphisms and on the XY chromosome were kept. Beta values were then normalized using the BMIQ method [52]. Batch effects at both the array and chip levels were corrected using the Combat algorithm [53].

The final discovery dataset contained 742,961 CpGs and 610 individuals (208 cases and 402 controls). The first validation dataset contained 478,687 CpGs across 279 samples. The second validation dataset contained 831,858 probes across 337 samples.

The final model used to identify differentially methylated positions (DMPs) was constructed using logistic regression, whereby phenotype (case or control) was considered the outcome variable and each CpG beta value and cell type proportion (natural killer cells, monocytes, B cells, CD8^+^ T cells, CD4^+^ T cells, and neutrophils) were considered predictors. For *CpG_i_*:$$glm\left(Outcome_{\frac{Case}{Control}}\;\sim \;CpG_{i}+Age+Sex+CellFraction+Region\;\right)$$

The mean difference in methylation between the case and control groups was used to assess the effect size (i.e., delta beta or Δβ).

When performing the genotype risk-corrected EWAS (correcting for known genotype risk loci (HLA-DRB1 RISK HAPLOTYPE) and aggregated polygenic risk score), we also incorporated genotype information in the model to identify genotype risk-independent DMPs:$$glm\left(Outcome_{\frac{Case}{Control}}\;\sim \;CpG_{i}+Age+Sex+CellFraction+Region+HLA\;Haplotype+PRS\right)$$

Significant DMRs were determined using multiple Fisher’s exact tests, Stouffer’s test, and the harmonic mean of the individual component FDRs, at *p* ≤ 0.05.

### 4.5. Sensitivity Analysis

Sensitivity analysis was performed on a range of covariates to assess their relative individual importance on CpG beta value. Numeric continuous variables were assessed using linear models such as $lm\left(CpG_{i}\sim Covariate\right)$ and categorical variables were assessed using ANOVA models such as $AOV\left(CpG_{i}\sim Covariate\right)$.

### 4.6. Genotype Analysis

Raw idat files were loaded in GenomeStudio (Illumina Inc., San Diego, CA, USA) for filtering. Recommendations from Guo et al., 2014 [54] were followed (except the ethnicity-specific steps) and the analysis-ready dataset was exported to plink [55] format. In short, QC was performed to filter haploid probes, then the probes were filtered based on their GenTrain score (score ≥ 0.7), cluster separation, SNP call rate (≥98%), and sample call frequency (≥95%). After exporting to plink format, samples were tested for gender mismatch, relatedness, and Hardy–Weinberg equilibrium. Plink was also used to generate association statistics, using Fisher’s exact test. The HLA-DRB1*15:01 haplotype was detected using the rs3135388 tag SNP [56].

For the CIT analysis, we used Haploview to identify the distribution of haplotypes. For each gene and group, we identified the haplotype that was most significant in an Χ^2^ test as a risk haplotype.

### 4.7. Scores and Receiver Operator Characteristic Curves

We constructed a genotype risk-corrected methylation score (grcMethScore) only incorporating DMPs identified in the genotype risk-corrected EWAS. The grcMethScore was constructed using the following formula:$${\mathrm{MethScore}}_{sample}=\sum \beta value_{\;at\;CpGi}\times \Delta \beta_{CpGi}$$
where $\beta value_{\;at\;CpGi}$ represents the beta value of this particular sample at a CpG of interest and $\Delta \beta_{CpGi}$ is a weight, using the value of delta beta identified in the genotype risk-independent EWAS (described in the methylation analysis Section 4.4).

For genotyping, we used two scores: the risk haplotype HLA-DRB1 score (either 0, 1, or 2) and a polygenic risk score (PRS) generated using PRSice [24] with all SNPs identified through the 2019 IMSGC GWAS [2].

ROC curves and the area under the curve (AUC) were computed using the R package ROCR [57] using the scores generated with logistic regression.
$$glm\left(Outcome_{\frac{Case}{Control}}\;\sim \;Predictor_{1}+\cdots +Predictor_{n}\right)$$
where outcome is the MS-Control outcome and *Predictors* are grcMethScore, PRS, cell proportion, sex, etc.

### 4.8. Immune Cell Deconvolution Analysis

Cell proportion estimates were obtained using the R package EpiDISH [58]. Cell proportions, in conjunction with M-values, were used to calculate cell type-specific DMPs, using a method inspired by the CellDMC function [25]. CellDMC has two major limitations. First, the results are biased based on initial cell proportions when using beta values. Second, CellDMC modeling is overburdened when incorporating numerous cell types, preventing the identification of effect in all or multiple cell types.

To compensate for this, we used a two-step approach using the combined dataset for maximum statistical power. First, we calculated cell proportion estimates using the R package EpiDISH. Second, we used the cell type proportions in conjunction with CpG methylation (M) values to calculate cell type-specific DMPs (csDMPs). Here we employed an adaptation of the functions in CellDMC [25]. Although it accounts for all cell types simultaneously, the base CellDMC regression model can become overburdened when incorporating many cell types (i.e., many terms), thus reducing the power and preventing the identification of cell-specific effects in multiple cell types.

Instead, for each cell type, we constructed a linear model as follows:$$lm\left(CpG_{i}\;\sim \;cellFrac+cellFrac:Phenotype+Covariates\right)$$
where *CpG_i_* represent the M-value at *CpG_i_*, *cellFrac* is the cell type fraction (ranging from 0 to 1), and *cellFrac: Phenotype* represents the interaction term between cell fraction and phenotype. CpGs with an interaction cellFrac:Phenotype estimate above 2 or under −2, as well as a *p*-value under 9.8 × 10^−8^, were kept for further downstream analysis.

### 4.9. Expression and DNA Methylation Analysis

Monocytes: Expression data were comprised of data from an independent, cross-sectional case–control study. Expression data from monocytes from previously published work were acquired from the authors under a data transfer agreement and DNA methylation data obtained using the 450 K platform were freely available from GEO [59] (GSE43976). Monocytes were isolated from PBMCs by magnetic separation (Miltenyi, Bergisch Gladbach, Germany) according to the manufacturer’s instructions and expression of HLA-DRB1 was quantified by real-time PCR, as previously described [16].

CD4^+^ T cells: Our group previously published the results of DNA methylation analysis using the 450 K array. From this study, we also had matched RNA available from 18 relapse-onset MS cases (unpublished results). The MS cases were all female and either treatment naïve or had been off MS-specific treatment for at least 6 months. They were matched with 33 age- and sex-matched non-MS controls.

The RNA from CD4^+^ T cells was extracted using the RNAeasy mini kit (Qiagen, Germantown, MD, USA) following the manufacturer’s protocols. The resulting RNA quality was assessed using the Agilent Bioanalyzer and met a quality cut-off of an RIN number of 7 or higher. Samples meeting QC requirements were hybridized to the Illumina HT12 expression array card (service provided by Diamantina Institute, Australia).

Expression analysis was carried using the R package beadarray [60]. In short, idat files were loaded using the readIdatFiles function, data were normalized using the normaliseIllumina function with the quantile method. Finally, the limmaDE function was used to determine differential gene expression and we used the illuminaHumanv4.db package to annotate the results.

B cells: The B cell dataset was composed of a total of 73 samples, including 23 controls (13 non-inflammatory neurologic disease controls and 10 healthy individuals) as well as 50 MS cases (39 RRMS and 11 SPMS). Peripheral blood mononuclear cells (PBMCs) were isolated directly after collection using standard Ficoll (GE Healthcare) and sodium citrate-containing preparation tubes (Becton Dickinson) procedures, respectively. CD19+ B cells were isolated using positive selection by magnetic separation on MACS MicroBeads (Miltenyi), according to the manufacturer’s instructions (>95% purity). Cells were prepared within an hour after sampling using an AutoMACS (Milteny Biotec, Bergisch, Germany) according to the manufacturer’s standard protocol and stored at −70 °C. DNA and RNA were extracted simultaneously using the Qiagen Allprep DNA/RNA kit (Qiagen, Venlo, The Netherlands) according to the manufacturer’s standard protocol and stored at −70 °C. The amount and quality of DNA were assessed using a NanoDrop ND-1000 spectrophotometer (NanoDrop Technologies Inc.). Samples with sufficient DNA amounts were used in further analysis. Processing of the samples for the Infinium HumanMethylationEPIC array (Illumina), including bisulfite conversion, was performed at the National Genomics Institutet (NGI) at the Science for Life Lab (Uppsala, Sweden).

RNA-Seq libraries were generated from 500 ng of total RNA using the Illumina TruSeq mRNA Stranded Library Preparation Kit (cat. no. RS-122-2103) according to the manufacturer’s protocol. Library quality was determined using the Agilent High Sensitivity DNA Kit (cat. no. 5067-4626) and a NanoDrop ND-1000 Spectrophotometer (NanoDrop Technologies Inc., Wilmington, DE, USA). Libraries were sequenced on the Illumina HiSeq 2500 as per the manufacturer’s instructions, and ∼20 M 75 bp paired-end reads were generated per sample. Read quality was assessed before and after trimming using FastQC v0.11.4. Illumina adapters, and low-quality nucleotides were trimmed using Cutadapt v1.9.1. Reads were aligned to the human genome (GRCh38), and read count per gene was determined using STAR.

All cell types:

Methylation profiles for every cell type were analyzed individually using the R packages Minfi [61] and ChAMP [50,51] following the pipeline according to Marabita et al. [62]. Briefly, type 1 and type 2 probes were normalized using quantile normalization and BMIQ. Sex was confirmed using the GetSex function in the Minfi package and cell type identity was confirmed using the cell type deconvolution method in the Minfi package based on the Houseman algorithm [63]. Age was confirmed using the AgeP function in the WateRmelon package [64]. Furthermore, sample identification was confirmed by running a 44 SNP panel at NGI, and genotypes were confirmed by overlapping SNP calls with data from our genetic studies. Except for data related to pwMS using rituximab and DMF because of the pairwise analysis, the following probes were filtered out: (i) probes not passing the detection cut-off *p*-value of 0.01, (ii) probes with known SNPs, and (iii) X and Y chromosome probes. Batch effects were identified using principal component analysis (PCA) and corrected using the ComBat function in the SVA package [53]. The loading of methylation profiles was performed in this manner for each dataset used in this study. DMPs were determined by linear modeling using the limma package [65] in a model that included age, sex, and B cell purity from the Houseman algorithm as covariates.

### 4.10. Correlation Analysis (DNA Methylation and Gene Expression)

Correlation analysis was performed using the R package corrplot. All three cell types were analyzed separately. In every case, a measure of expression (from RT-PCR, array, or sequencing) was correlated with a beta value for every sample. The Pearson R coefficient is reported on the plots.

### 4.11. metQTL Analysis

We tested for the influence of genotype on methylation in two instances. Each time, we performed an ANOVA test:$$AOV\left(CpG\;\sim \;SNP_{AA-AB-BB}\right)$$
where CpG represents the beta value at a specific CpG and $SNP_{AA-AB-BB}$ represents the genotype group.

We tested both the influence of the *HLA-DRB1*15:01* risk haplotype on genome-wide methylation and all SNPs within ±18 kBp (36 kBp window) for each DMP identified in the genotype risk-independent EWAS.

### 4.12. Over-Representation Analysis

ORA was performed using the online Reactome platform [66] to analyze various gene lists with the inclusion of IntAct [67] interactors in the analysis. Pathways were kept if the FDR value of the individual component was under 0.05. Figures were generated using the R package ggplot2 [68].

## 5. Conclusions

This study provides compelling evidence that DNA methylation, independently of genotype, is strongly modified in pwMS and is implicated in the NOTCH and axon guidance pathways. The results of this study advance previous work implicating B cells and monocytes in MS pathogenesis by revealing that differential methylation is present at disease onset. The effect of differential methylation of B cells and monocytes is linked to gene expression differences in these cell types. While therapies affecting B cells have been shown to be extremely effective in MS, monocytes may also represent a future therapeutic target. Finally, we show that methylation, after correcting for known genetic risk, outperforms genetics when discriminating between MS and non-MS individuals (see Figure 3). This provides compelling evidence that environmental exposure and lifestyle play a larger role in MS onset than genetics.

## Figures and Tables

**Figure 1 ijms-24-12576-f001:**
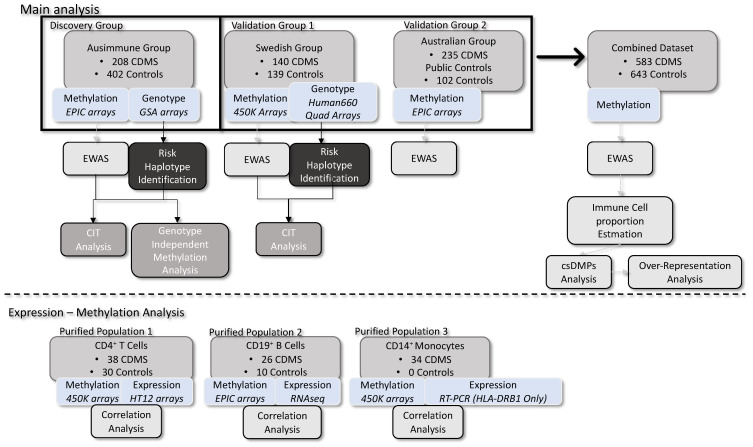
Study design and workflow.

**Figure 2 ijms-24-12576-f002:**
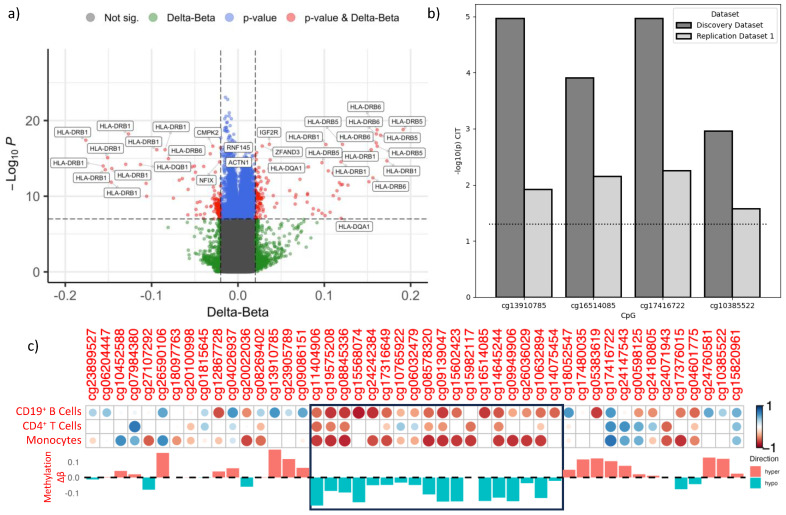
Methylation differences at the MHC class II locus are the largest between controls and pwMS. (**a**) Combined analysis of all three cohorts identified DMPs associated with MS. Volcano plot representing the 452,453 CpGs analyzed in the combined cohort DMP analysis. Red dots indicate DMPs that passed both the adjusted *p*-value (−log10(*p*) ≤ 7) and delta beta (±0.02) thresholds. Blue dots represent DMPs that passed the *p*-value threshold only and green dots indicate DMPs that passed the delta beta threshold only. The locus to which each DMP is mapped is indicated for genes that passed both the *p*-value and delta beta thresholds. (**b**) Genotype at *HLA-DRB1* mediates methylation. Results of the CIT analysis measuring haplotype-related mediation of MS risk through methylation. p CIT: overall Omnibus CIT *p*-values. Dotted horizontal line represents significance threshold of *p* ≤ 0.05. The CpGs used for *HLA-DRB1* are indexed CpGs for each of the 4 DMRs associated with MS. Dark gray = discovery dataset, light gray = validation dataset 1. (**c**) Correlation between *HLA-DRB1* expression and CpG methylation. Top panel: cell types are listed in rows, CpGs are in columns, except for column 1, which represents gene-of-interest expression in each cell type. The correlation coefficient (R) is represented through circle size and color. A larger circle area indicates a high absolute R value (maximum 1). Red indicates negative values; blue indicates positive values. The black rectangle represents the hypomethylated DMR 2 at HLA-DRB1 identified in whole blood and cell-specific analysis. Bottom panel: whole blood methylation differences (Δβ) between pwMS and controls. Pink represent hypermethylation and blue represents hypomethylation.

**Figure 3 ijms-24-12576-f003:**
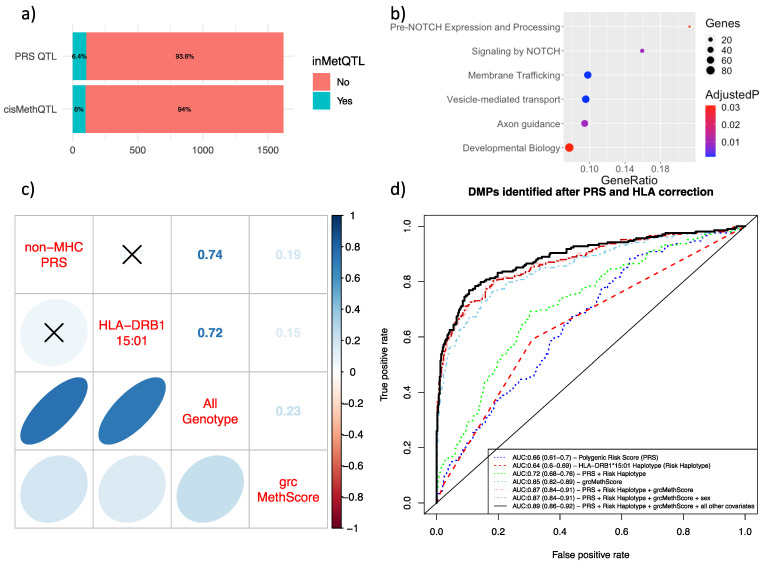
Methylation analysis correcting for known MS risk loci. (**a**) Proportion of DMPs mediated by polymorphisms. Either close polymorphisms (cis methQTLs within ± 20 kb) or SNPs that were used to construct the PRS (PRS QTL). (**b**) ORA on all DMPs identified in the genotype-corrected EWAS. Colors represent FDR *p*-values, dot size represents the number of genes, and gene ratio represents the number of genes in our list divided by the total number of genes in the given pathway. (**c**) Correlation matrix between various scores. Crossed cells represent non-significant values. PRS = PRS inferred from 201 non-MHC SNPs. DR15 haplotype = risk haplotype conferring 3× risk of MS, HLA-DRB1*15:01. All genotype = risk score incorporating PRS and HLA-DRB1*1 5:01. Genotype risk-corrected. MethScore = methylation score corrected for genotype at known MS risk loci. (**d**) ROC curves related to each score discriminating between MS and non-MS controls. AUC = area under the curve.

**Figure 4 ijms-24-12576-f004:**
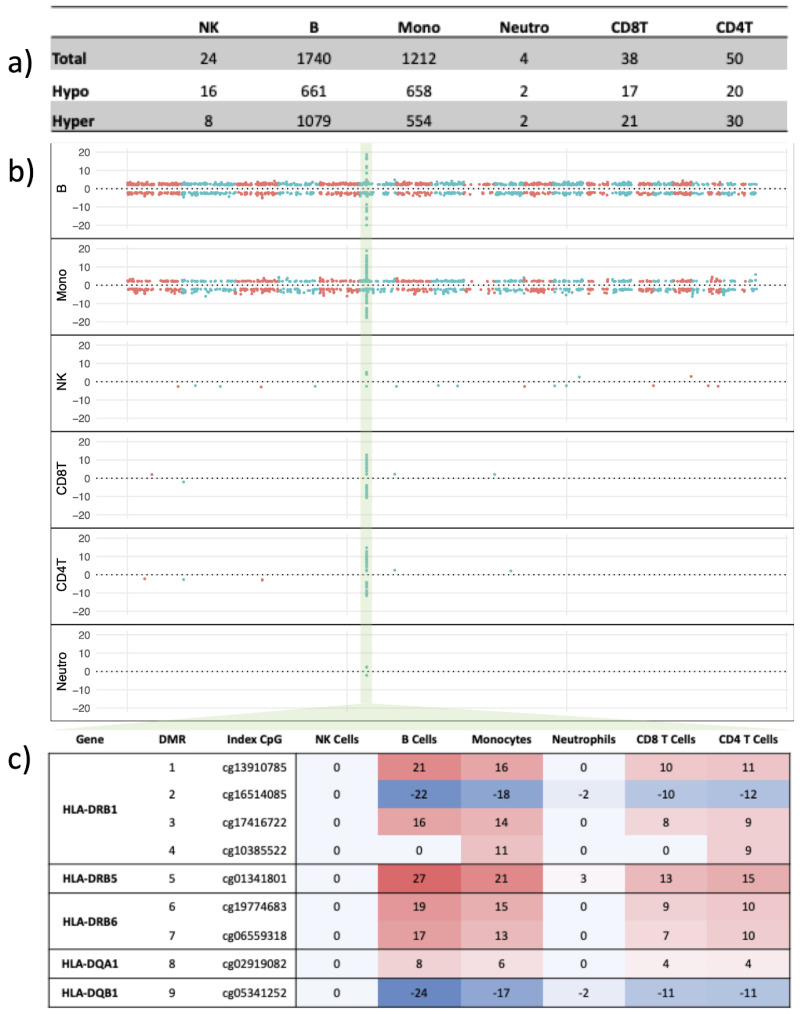
Genome-wide cell-specific DNA methylation profile. (**a**) Total number of cell type-specific DMPs (csDMPs) and breakdown by hyper- and hypomethylated. (**b**) Plot representing genome-wide effect size for all csDMPs, by cell type. Each dot represents a significant csDMP. The *Y*-axis represents the model estimate for effect size and the *X*-axis represents genome coordinates; chromosomes are in chronological order and represented by alternating color. The highlighted region represents the MHC class II regions (**c**) MHC class II region cell-specific DNA methylation. Shown is the effect size in each immune cell type for the index CpG of the nine major DMRs identified in whole blood. Values are estimates extracted from our cell-specific models. Blue/negative values represent negative estimates or hypomethylation at those CpGs, while red/positives values represent positive estimates or hypermethylation. A darker color indicates a higher effect size.

**Figure 5 ijms-24-12576-f005:**
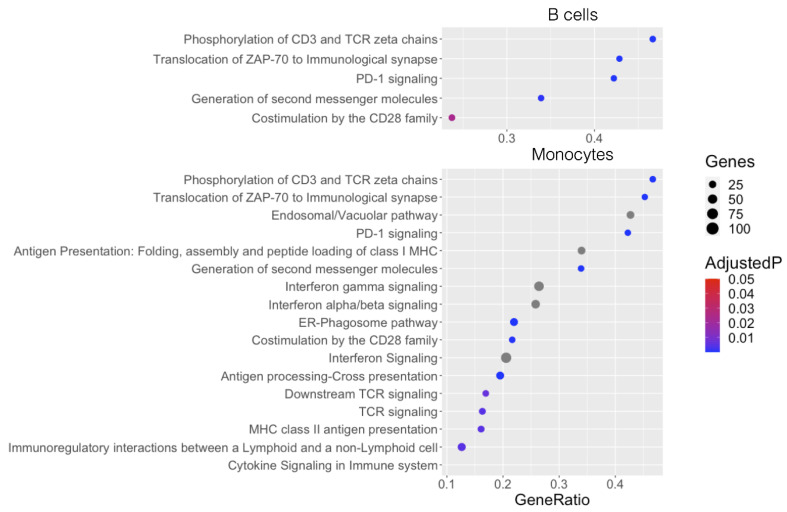
ORA of genes containing cell type-specific DMPs (csDMPs). Lollipop plots represent the ORA results using gene lists of the csDMPs attributed to B cells and monocytes. The size of the marker represents the number of genes. The *X*-axis represents the gene ratio or the number of genes differentially methylated divided by the number of all genes in the given pathway. Levels of significance are represented by colors.

**Figure 6 ijms-24-12576-f006:**
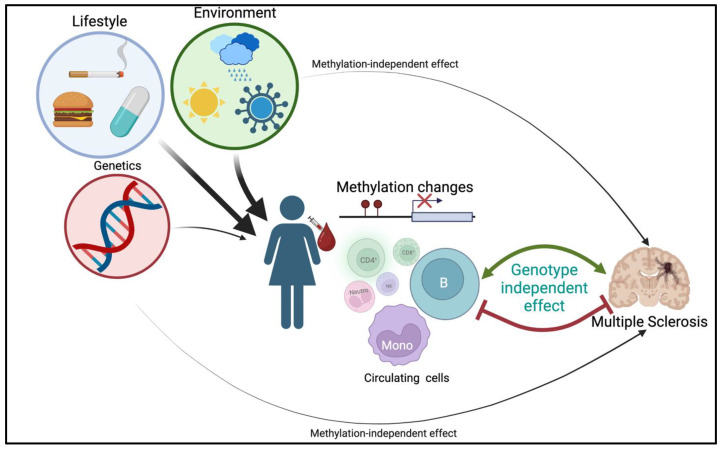
Diagram showing the influence of lifestyle, the environment, and genetics on methylation changes in specific immune cell types in MS.

## Data Availability

The datasets generated and analyzed during the current study are not publicly available due to the General Data Protection Regulation but are available from the corresponding author upon reasonable request.

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
