# Peer review of "DNA Methylation Signatures of Multiple Sclerosis Occur Independently of Known Genetic Risk and Are Primarily Attributed to B Cells and Monocytes"

_ijms, 2023, doi:10.3390/ijms241612576_

Round 1

Reviewer 1 Report

In this study, the authors evaluate if differential DNA methylation can strongly differentiate risk factor for MS as opposed to the conventional genetic risk loci. Overall, the authors show that DNA methylation is a robust indicator of disease and is independent of genetic risk haplotype. They employ a corrected risk score to account for changes in genotype and find no correlation between differentially methylated regions and SNPs that are associated in the genetic risk haplotype. The data is still correlative and genome wide analysis is always harder to evaluate in terms of disease predictions, but the authors make a good case for their claims. The data presentation is slightly hard to grasp. Supplementary documents appear to have a ton of information and text which is unusual. I have some minor comments below:

1. CD4+ cells are significantly increased in the MS group but they don't really have a change in gene expression in the HLA-DRB haplotype which is consistent with the moderate set of csDMPs. In terms of cell count, the monocyte and B cell population numbers are not different but they have the highest amount of csDMPs reported. This seems intriguing. There must be some cell expression changes for the differences in cell count although it is quite small. Can the authors account for why one cell type is more susceptible than the other? Do the authors think that other epigenetic changes are happening in the CD4+, NK cells?

2. I am also curious whether in this dataset the authors also found the 79 DMPs that they previously report for CD8+T cells in their 2015 Clinic. Epigenetics paper? Are DMPs occurring at certain areas of the genome or is it different and depends on the datasets? Are DMPs in different patients occurring at different regions of the genome? 

2. One point the authors bring up, which I understand is hard to address is whether the methylation changes occur as a result or is a cause of MS. Even in the early group, the disease has already started. Risk factors that predispose you to the disease may still be completely genetic and acting upstream of DNA methylation changes. Regardless, this study highlights the need to also focus on epigenetic changes in MS patients.

3. There appears to be a disproportionate amount of data in supplementary files while the main figures are pretty scant. This makes it really hard to focus on the paper's findings. The near perfect correlation of DNA methylation and HLA-DRB in the HLA-association could be included in the main figure. Clearly  that is important information. Also, the work flow in figure 5 seems to be really important to understand what the discovery, replicate and combined data set mean. Why is that hidden at the end of the text? For these studies to be accessible to open readership, the method and work flow should be initially detailed. 

Reviewer 2 Report

In this manuscript, Xavier and colleagues state that the DNA methylation signatures of multiple sclerosis occur regardless of known genetic risk and are primarily attributed to B cells and monocytes.

The idea is interesting but the research design is inappropriate.

The authors performed a methylation matrix of whole blood from MS patients identifying 3,218 differentially methylated locations (DMPs). 13 of 18 DMPs, which overlapped in both replication groups with the same directionality, localized to the HLA-DRB1, HLA-DRB5 or HLA-DRB6 genes, however with different frequencies.

Here arises the first experimental problem, i.e. only 13 DMPs out of 3,218 overlapped probably because most of the interesting positions were diluted in the analyzed cell population which was heterogeneous.

Furthermore, the authors correlated the gene expression of isolated cells with the methylation of 13 DMPs identified from whole blood. This is indirect and random data. The expression of isolated cells could depend on multiple variables, how can the authors claim that there is a causal link?

Finally, the same authors state that the high clinical heterogeneity and I would add of the experimental design, represents a limitation of the work that does not allow reaching conclusions unsupported by the results shown.

The authors should directly confirm the correlation between expression and methylation data on patients' B lymphocytes and monocytes.

Reviewer 3 Report

This manuscript by Xavier et al. describe array based (EPIC/450K) genome wide DNA methylation analysis of cases of early stages of multiple sclerosis (and controls), the authors found a unique epigenetic signature of DNAm in MS, verified their findings based on previous studies and found that most of the changes in DNAm arise from B cells and monocytes. This is an interesting manuscript and below are my comments which will be helpful for the authors.

1.     Sex was analyzed as co-variate. Autoimmune disease tend to affect woman more than man, so it would be nice to run the analyses (at leaset the major ones) separately for males and females.

2.     A delta beta of 2% was used for the cutoff which seems to be low, this should be explained. I encourage the authors to also analysis CpGs with a larger delta beta cutoff as well. Also, since some of the analysis is cell-type specific, one would be expected higher differences in methylation as cell variability is not a confounder in this case.

3.     Over-representation analyses (fig1 , Fig 3, and in supp data) were done on all differentially methylated  CpGs regardless of a) the genomic position of the CpG (promoter, gene body) and direction of the change (hypo vs hyper) these factors affect the biological effect of methylation on gene expression. Therefore, it does not provide much information to show the ORA analysis as they stand. I believe breaking the analysis for hypo / hyper ( or use the directionally as a value to analyze activation/inhibition of a pathway as done by tools such as Ingenuity Pathway Analysis) would be better. Same for analyzing separately for differential CpGs within promotes / gene bodies / distal sites)

4.     Are the cell-type specific effects (mostly differential CpGs  in b cells and monocytes) holds also for each study alone or only on the aggregated data?

5.     Minor: I don’t think the other studies can be called replication since they were done before the current study. I think it is more accurate to refer to them as validations or the like.

6.     The expression data is shown in the Supp file. But there is no reference to it in the main file, besides a short discussion (had to look for very long time to find it in the supp data).
Then, the reader gets to the method section where gene-expression analysis is explained in details without realizing where is the data shown in the text. Please fix it, as it stand it is very unclear.

7.      

Round 2

Reviewer 2 Report

I really appreciate the authors' efforts in responding to the requests. However, the data is heterogeneous and the results do not support the conclusions.

Author Response

We thanks the reviewer for their comments.

However, we disagree with their statement. 

Our main claims are supported by the data:

1) HLA is the main source of methylation variance and is driven by genotype -> supported by methylation / expression and genotype data

2) Genotype-independant methylation is a better discriminator of MS. -> Supported by our genotype-independant analysis

3) Methylation differences are strongest in Monocytes and B cells -> supported by our cell-specific analysis.

Reviewer 3 Report

The authors answered all my comments and concerns. Thank you

Author Response

We thanks the reviewer for their comments.